# Is Integrated Management of *Gymnandrosoma aurantianum* Possible with *Trichogramma atopovirilia* and Novel Products Used in Citrus Orchards in Brazil?

**DOI:** 10.3390/insects14050419

**Published:** 2023-04-27

**Authors:** Lucas Vinicius Cantori, Fernando Henrique Iost Filho, Juliano de Bastos Pazini, Alexandre José Ferreira Diniz, Pedro Takao Yamamoto, José Roberto Postali Parra

**Affiliations:** Department of Entomology and Acarology, “Luiz de Queiroz” College of Agriculture, University of São Paulo (USP), 11 Pádua Dias Ave, Piracicaba 13418-900, Brazil; fernandohiost@usp.br (F.H.I.F.); julianopazini@usp.br (J.d.B.P.); aleniz@usp.br (A.J.F.D.); pedro.yamamoto@usp.br (P.T.Y.); jrpparra@usp.br (J.R.P.P.)

**Keywords:** egg parasitoid, citrus fruit borer, integrated pest management, lethal effect, sublethal effect, toxicity

## Abstract

**Simple Summary:**

The citrus fruit borer has become one of the main pests in citrus orchards in Brazil, possibly due to biological imbalances caused by multiple insecticide applications. This pest is managed with insecticides, but studies indicate a potential use of *Trichogramma atopovirilia* to control its eggs. In this way, we evaluated the selectivity of new products that could eventually be used in the Brazilian citrus orchards on *T. atopovirilia*. Among the products tested, spinetoram was considered harmful. Cyantraniliprole, cyantraniliprole + abamectin, abamectin, sulfoxaflor, and the entomopathogenic fungi *Cordyceps fumosorosea* were considered selective and non-persistent to the parasitoid.

**Abstract:**

In Brazil, the citrus fruit borer, *Gymnandrosoma aurantianum* Lima, 1927, is a serious pest in orange orchards, causing an annual loss of 80 million US dollars, and is managed with multiple insecticide applications, often 56 in a single season. On the other hand, the parasitoid wasp *Trichogramma atopovirilia* Oatman & Platner, 1983 has the potential for controlling *G. aurantianum* by attacking its eggs. Considering the intensive insecticide applications in citrus orchards in Brazil to control the large complex of pests, especially *Diaphorina citri* Kuwayama, 1908, evaluation of the harmful effects of insecticides on *T. atopovirilia* is important to maximize efficiency in managing *G. aurantianum*. Here, we tested the effects of new products used in citrus orchards (cyantraniliprole, cyantraniliprole + abamectin, abamectin, sulfoxaflor, spinetoram, flupyradifure, and *Cordyceps fumosorosea* (Wize) Kepler, B. Shrestha & Spatafora) on adults and pupae of *T. atopovirilia*. Of the insecticides tested, spinetoram caused the highest impacts on *T. atopovirilia* parasitism, longevity, emergence, and mortality. The other products caused more sublethal than lethal effects and were classified as 1 and/or 2 in the IOBC/WPRS classification. Abamectin, cyantraniliprole, cyantraniliprole + abamectin, and the entomopathogenic fungus *C. fumosorosea* were classified as short-lived. Except for spinetoram, these products were classified as selective. In this study, spinetoram was considered harmful to *T. atopovirilia* and, therefore, should be managed carefully in IPM programs combining this parasitoid. In order to safely use this insecticide, one should respect the interval of release of the parasitoid, which is 21 days after its spraying. The novel products tested, cyantraniliprole, cyantraniliprole + abamectin, abamectin, sulfoxaflor, and the entomopathogenic fungi *C. fumosorosea* were selective and non-persistent to *T. atopovirilia*. These products are possible replacements for non-selective insecticides to achieve higher control from both chemical and biological tools.

## 1. Introduction

The use of biological control agents and their conservation in an agricultural system are among the integrated pest management (IPM) strategies to maintain pest populations below the level of economic damage [1,2]. Considering that, in many cases, pest control requires a simultaneous application of biological control agents and insecticides, the success of IPM programs may depend on the use of selective phytosanitary products, i.e., those that efficiently act against insect pests present in an agroecosystem and that have the least possible impact on non-target organisms [2,3].

Although chemical control is used most often worldwide, agrochemicals can significantly reduce the efficiency of predators and parasitoids, for example, from the genus *Trichogramma* [2], which is currently used in pest control programs on approximately 4 million hectares in Brazil [4]. The harmful effects of agrochemicals on natural enemies can cause mortality or biological disorders that will impact the fitness of a particular individual and its offspring. The use of compatible products is essential in the release of these parasitoids [2]. In general, these natural enemies are more affected by insecticides than by other agrochemicals, as they can cause high adult mortality, in some cases up to 30 days after spraying [5,6,7].

Currently in Brazil, the citrus fruit borer, *Gymnandrosoma aurantianum* Lima, 1927 (Lepidoptera: Tortricidae), is one of the main pests in orange orchards. This micro lepidopteran is managed with insecticides that are often applied on a scheduled basis. The citrus fruit borer became a key pest for Brazilian citrus orchards in the late 1980s, possibly due to biological imbalances caused by the excessive application of agrochemicals to control the vectors of citrus variegated chlorosis (CVC) [8].

A research program led by the Department of Entomology and Acarology at ESALQ/USP, with the support of Fundecitrus (Fund for Citrus Protection, Brazil) and the participation of UFV (Federal University of Viçosa, Brazil) and the University of California, Davis (CA, USA), developed a technological package, including the identification of a sex pheromone for citrus fruit borer adults. The product, patented as Ferocitrus Furão^®^, has been successfully used since the end of 2001 and prevented losses of more than 1.3 billion dollars in the 2001–2012 period [9]. However, after 2004 when “greening” (HLB) was first recorded in Brazil, resulting in massive amounts of agrochemicals applied to control the vector *Diaphorina citri* Kuwayama, 1908 (Hemiptera: Psyllidae), a new population explosion of *G. aurantianum* occurred in the citrus-growing areas of São Paulo state [10].

Generally, *G. aurantianum* females lay one egg per fruit, and after hatching, the caterpillar penetrates the fruit, making it unfeasible for immediate consumption and industrial processing [10]. Currently in Brazil, even though *G. aurantianum* is managed with weekly applications of insecticides on a prescribed schedule, the estimated annual loss is on the order of 80 million dollars because of the damage caused by the citrus fruit borer. This means that about 30 million boxes of oranges are annually lost, about equal to the losses caused by “greening” [10].

On the other hand, laboratory and field studies have demonstrated the potential of using the parasitoid wasp *Trichogramma atopovirilia* Oatman & Platner, 1983 (Hymenoptera: Trichogrammatidae) to control *G. aurantianum* by parasitizing its eggs [11,12,13,14]. Therefore, considering the high dependence on agrochemicals in citrus orchards in Brazil to control the large complex of pests, mainly *Diaphorina citri*, tests to evaluate the harmful effect of insecticides on *T. atopovirilia* are essential to maximizing the efficiency of managing *G. aurantianum* through the harmonious integration of chemical and biological controls. We evaluated the effects of insecticides on *T. atopovirilia* adults and pupae, their impacts on parasitism capacity, and their biological persistence. This information will make it possible to select insecticides with the lowest toxicity to the natural enemy and to adjust the spraying schedule to release parasitoids at the appropriate time to minimize their exposure to insecticide residues.

## 2. Materials and Methods

### 2.1. Insect Rearing

Orange fruits attacked by *Gymnandrosoma aurantianum* were collected from citrus orchards in different municipalities of São Paulo state (20°33′33″ S, 48°34′8″ W (Barretos), 20°56′59″ S, 48°28′44″ W (Bebedouro), 20°54′26″ S, 48°38′29″ W (Monte Azul Paulista), 21°35′46″ S, 48°48′48″ W (Itápolis), 21°3′60″ S, and 48°24′32″ W (Taquaral)), and brought to the Insect Biology Laboratory of the Department of Entomology and Acarology (LEA), ESALQ/USP. The fruits were sliced open, and the larvae of the citrus fruit borer were removed and transferred to glass tubes (2.5 cm wide × 8.5 cm high, the size used in all procedures) with an artificial diet [15], where they matured to adults. After emergence, the adults were transferred to PVC cages (15 cm wide × 15 cm high) that were lined inside with white paper, where the eggs were collected [16]. The colony was maintained at 26 ± 2 °C, 50 ± 10% RH, and 14:10 L:D photoperiod. 

*Trichogramma atopovirilia* individuals were collected in eggs of *Helicoverpa zea* (Boddie, 1850) (Lepidoptera: Noctuidae), in a corn-growing area in São José dos Pinhais, Paraná, Brazil (25°32′06″ S, 49°12′21″ W, elevation 906 m) (temperate climate with mild summers, Köppen C*fb*). The wasps were maintained for more than 20 generations (the life cycle (egg-adult) of this parasitoid is about 10 days [11,12,13]), on the eggs of *Ephestia kuehniella* (Zeller, 1879) (Lepidoptera: Pyralidae) in a rearing room in the same environmental conditions as above, at the Insect Biology Laboratory [17].

### 2.2. Insecticides

The insecticides were selected according to the trend of Brazilian citrus market requirements. For example, the application of organophosphates and pyrethroids to citrus makes it difficult to export fruit and juice, mainly to Europe and the U.S.A. These markets are increasingly requiring attention to the residues of these products. Therefore, we opted for products that will likely be used in the citrus industry soon, following the constantly changing list of citrus protection products, Protecitrus, maintained by Fundecitrus (https://www.fundecitrus.com.br/Protecitrus accessed on 10 February 2020) (Table 1).

### 2.3. Evaluation of Lethal and Sublethal Effects on Adults of T. atopovirilia

Pieces of paper containing 25 unparasitized eggs of *G. aurantianum* up to 24 h old were immersed in the insecticide solutions for 5 s [18]. These pieces of paper were then dried and kept together with a previously mated *T. atopovirilia* female for 24 h in test tubes containing a drop of pure honey and covered with a piece of plastic film to allow parasitism. After 24 h, the females were offered ≈200 *E. kuehniella* eggs to parasitize, which were changed daily until the female died. This experiment had 30 females per treatment, each considered a replication. The females were maintained in a climate-controlled room with a temperature of 25 ± 1 °C, 14:10 (L:D) photoperiod, and relative humidity of 60 ± 10%.

Mortality, parasitism capacity, and longevity of the surviving females were evaluated. Parasitism was evaluated based on the color of the eggs of *E. kuehniella*, either black (parasitized) or yellowish white (non-parasitized), observed under a stereoscopic microscope [19]. The percentage of emergence (number of eggs with holes in relation to the number of black eggs) and sex ratio—the proportion of females in the population (based on the adults’ antennae)—were also evaluated [20].

The effects on offspring were evaluated based on the F1 parasitism capacity for three days (number of *E. kuehniella* eggs parasitized by each 24-h-old female wasp, previously mated, for three days) replacing *E. kuehniella* eggs daily. Thus, the emergence of F2 offspring was also evaluated. The assays of sublethal effects on the offspring were carried out with 10 replicates of each treatment, each female representing one replicate.

The insecticides evaluated were classified into toxicological categories based on their effects on reducing the beneficial capacity of the parasitoid (parasitism), according to the IOBC/WPRS recommendations: class 1 = innocuous (<30% reduction), class 2 = slightly toxic (30% to 80% reduction), class 3 = moderately toxic (>80% to 99% reduction), and class 4 = toxic (>99% emergence reduction) [21].

### 2.4. Evaluation of Lethal and Sublethal Effects on Pupae of T. atopovirilia

The lethal effects of insecticides on the pupae (7 days after parasitism) of *T. atopovirilia* were evaluated by observing the emergence of adults. Pieces of paper containing 25 eggs of *G. aurantianum* were submitted to parasitism for 24 h, and then (7 days after) were immersed in an insecticide solution for 5 s [18]. After drying, the eggs were placed in glass tubes and covered with plastic film for the emergence of adults. The assay used 30 replicates per treatment. The pupae were maintained in a climate-controlled room with a temperature of 25 ± 1 °C, 14:10 (L:D) photoperiod, and relative humidity of 60 ± 10%. The emergence rate, longevity, and sex ratio of adults were evaluated. The toxicity of the insecticides, in terms of the reduction in emergence, was classified according to the above IOBC/WPRS recommendations [21].

### 2.5. Residual Activity (Persistence) of Insecticides to T. atopovirilia

Two-year-old orange seedling of *Citrus sinensis* (L.) Osbeck (Rutacea) of the Valencia variety were sprayed with the selected insecticides (Table 1), with a volume corresponding to ≈250 mL per orange seedling, using a Jacto PJH manual sprayer (Jacto do Brasil S.A., Pompéia, SP, Brazil) equipped with an FL-5VS conical nozzle (Teejet Technologies Company, São Paulo, SP, Brazil) [22] and maintained in a greenhouse under natural ambient temperature and partial shading. The pots containing the orange seedlings were irrigated, only in the soil, every two days. To provide enough leaves, each treatment had three orange seedlings. The control treatment was sprayed with water. 

At 3, 7, 14, 21, and 30 days after the insecticides were sprayed (DAS), mature leaves from each treatment were collected and transported to the laboratory [23]. Each treatment had 20 replicates, each consisting of a 24-h-old female of *T. atopovirilia* and a leaf treated with the respective insecticide placed in a glass tube with a drop of pure honey and covered with plastic film. The females were maintained in a climate-controlled room with a temperature of 25 ± 1 °C, 14:10 (L:D) photoperiod, and relative humidity of 60 ± 10%. Mortality was assessed 24 h after the parasitoids were exposed to the treated leaves. If a product reduced insect mortality by less than 30% compared with the control treatment, it was classified according to the IOBC/WPRS persistence scale, as 1—short-lived (<5 days), 2—slightly persistent (5–15 days), 3—moderately persistent (16–30 days), or 4—persistent (>30 days) [21]. For the surviving females, *E. kuehniella* eggs were offered and replaced daily for three days to evaluate the parasitism.

### 2.6. Statistical Analysis

Data residuals were tested for Shapiro–Wilk normality and Bartlett variance homoscedasticity. Because the residuals did not follow a normal distribution, the data were submitted to nonparametric tests. Longevity and parasitism data were subjected to a Kruskal–Wallis test and compared by Dunn’s test with Bonferroni correction (*p* < 0.05). Emergence and sex-ratio data were compared using a generalized linear model, binomial, or quasi-binomial, depending on the best fit for each data set (*p* < 0.05). The analyses were carried out with the programming language R version 4.0 (R Development Core Team, 2017).

## 3. Results

### 3.1. Evaluation of Lethal and Sublethal Effects on Adults of T. atopovirilia

The results indicated significant differences in mortality rates of *T. atopovirilia* exposed to eggs treated with insecticides (χ^2^ = 36.56, df = 7, *p* < 0.001) (Table 2). Adult mortality was highest when eggs treated with spinetoram were offered, while the other products caused <10% of *T. atopovirilia* adults.

*T. atopovirilia* females showed significant decreases in the percentage of *G. aurantianum* eggs parasitized in 24 h (χ^2^ = 26.99, df = 7, *p* < 0.001) (Table 2). The 24-h parasitism of treated *G. aurantianum* eggs ranged from 3.71 (abamectin) to 9.36 eggs (cyantraniliprole) (Table 2). The reduction in parasitism was highest in spinetoram (77.31%), abamectin (53.81%), and cyantraniliprole + abamectin (45.53%), followed by flupyradifurone (27.62%). The products that least affected the parasitism were *C. fumosorosea* (3.57%), sulfoxaflor (0%), and cyantraniliprole (0%) (Table 2).

The other parameters, total parasitism (χ^2^ = 49.91, df = 7, *p* < 0.001) and longevity of females (χ^2^ = 54.62, df = 7, *p* < 0.001), were similar for all insecticides tested, except for spinetoram. Spinetoram reduced total parasitism and female longevity the most. Cyantraniliprole, flupyradifurone, sulfoxaflor, and *C. fumosorosea* reduced total parasitism and female longevity the least (Table 2).

The longevity (χ^2^ = 24.85, df = 7, *p* < 0.001) and emergence of the offspring (χ^2^ = 27.17, df = 7, *p* < 0.001) were most affected by the insecticides abamectin + cyantraniliprole, abamectin, and spinetoram. The emergence of individuals from treated *G. aurantianum* eggs was most affected by spinetoram (12%), statistically differing from the other treatments (Table 3). Despite the effects on the F0 female parasitism caused by the products, the parasitism of females (F1) was not affected by the insecticides (Table 3).

The sex ratio (proportion of females in the population) was most affected by cyantraniliprole and cyantraniliprole + abamectin, 0.21 and 0.28, respectively (χ^2^ = 99.89, df = 7, *p* < 0.001) (Table 3). These numbers are far below the expected ratios for quality checking in a population of parasitoids [24].

### 3.2. Evaluation of Lethal and Sublethal Effects on Pupae of T. atopovirilia

Among the insecticides tested, only spinetoram drastically reduced (92.92%) the emergence of *T. atopovirilia* when applied to 7-day-old parasitoid pupae in *G. aurantianum* eggs (χ^2^ = 79.75, df = 7, *p* < 0.001). The other insecticides reduced emergence by less than 5% and were considered innocuous according to the IOBC/WPRS classification (Table 4).

### 3.3. Residual Effects (Persistence) of Insecticides on T. atopovirilia

Based on the results, abamectin, cyantraniliprole, cyantraniliprole + abamectin, and *C. fumosorosea* were classified as short-lived, that is, *T. atopovirilia* can be released only 5 days after application of these products. Sulfoxaflor and flupyradifurone were classified as mildly persistent because they affect the parasitoid up to 15 days after application. Spinetoram was classified as moderately persistent, as it killed over 30% of the parasitoids two weeks after application (Figure 1).

The number of eggs parasitized by females in contact with *Citrus* leaves 3 DAS ranged from 9.16 (abamectin) to 33.56 (flupyradifurone) (χ^2^ = 21.46, df = 7, *p* = 0.0015), and emergence was up to 88% (χ^2^ = 69.55, df = 7, *p* = 0.008). On days 3, 7, and 10, there were insufficient replicates to analyze because of the high mortality caused by spinetoram (Table 5).

## 4. Discussion

In this study, we evaluated the compatibility of new products that will be more widely used in Brazilian citrus crops with *T. atopovirilia*, a parasitoid that will likely be commercially used for citrus fruit borer control. Among the insecticides tested, spinetoram caused the highest parasitoid mortality. Takahashi [25] found similar results when eggs of the noctuid lepidopterans *Anticarsia gemmatalis* Hübner, 1818 and *Chrysodeixis includens* (Walker, 1858) treated with spinetoram were offered to *T. atopovirilia.* Takahashi [25] observed 100% mortality after 24 h in both hosts, similar to the findings of Khan [26] for *Trichogramma chilonis* Ishii, 1941 (Hymenoptera: Trichogrammatidae). According to the results of the present study and the results obtained by Takahashi [25] and Khan [26], spinetoram is lethal to adults of several species of *Trichogramma*, independent of the surface characteristics of the host eggs. Spinetoram may cause mortality in two ways: first, by the contact of the tarsus with the egg surface, and second, by the contact with the ovipositor as it penetrates during the parasitism process. This insecticide may penetrate through the egg aerophiles and, regardless of its host species’ morphology, contaminate the parasitoid [27]. It is well documented that some spinosyns are harmless to predators but often negativity affect hymenopteran parasitoids [28]. The selective property of spinosyns is quite controversial. The use of insecticides derived from spinosyns is sometimes recommended in IPM programs because of their demonstrated low toxicity for non-target organisms in field conditions. However, in other cases, spinosyns are harmful to biological agents [28]. Although, in the present study, spinetoram reduced parasitoid emergence in laboratory conditions by almost 100%, further work is needed to determine its toxicity to *T. atopovirilia* in the field.

Cyantraniliprole has been reported to kill more than 80% of *Trichogramma* adults exposed to treated eggs in the first 24 h [29]. In our study, this product caused 3.33% mortality in the first 24 h. Although cyantraniliprole is a novel insecticide of the diamide group and seems very efficient for insect pest control, the mortality of egg parasitoids cannot be overlooked in IPM programs. In the present study, the same active ingredient was classified as non-persistent.

Based on IOBC standards, flupyradifurone is often classified as harmless to *Trichogramma* after a 24-h exposure [30]. Iost Filho et al. [31] and Costa et al. [32] found that flupyradifurone did not affect the emergence and parasitism rate of *Trichogramma pretiosum* Riley, 1879. In contrast, based on the results of the current study, we classified flupyradifurone as slightly persistent, causing 50% mortality 10 days after the citrus leaves were sprayed. We found an incongruence, because flupyradifurone caused zero mortality when applied to *G. aurantianum* egg surfaces after 24 h but caused about 60% mortality 3 days after the citrus leaves were sprayed. This also occurred with other products, where mortality was higher at the beginning of exposure to citrus leaves than to *G. aurantianum* eggs. This may have occurred because the leaf’s surface is larger than the surface of the egg, hence the parasitoids’ exposure to the products was higher. These results show the importance of the persistence test, which is closer to field reality, and would prevent the recommendation of a product that would be more lethal by adopting the method of offering treated eggs in test tubes. Also, the interaction of the insecticide flupyradifurone on citrus plants may have increased the toxicity of the product, as it is a systemic product. This interaction does not occur with this pest’s eggs [33].

Parasitism 24 h after exposure to treated *G. aurantianum* eggs ranged from 3.71 (abamectin) to 9.36 eggs (cyantraniliprole). The reduction in parasitism was highest in spinetoram (77.31%), abamectin (53.81%), and cyantraniliprole + abamectin (45.53%), followed by flupyradifurone (27.62%). Another study showed that parasitism by *T. atopovirilia* is twice as high in eggs of *Diaphania hyalinata* (L., 1767) (Lepidoptera, Pyralidae) and emergence is three times as high when treated with abamectin [34] than the treated *G. aurantianum* eggs found here. This might be explained by the difference in egg surfaces between the host species. The present study is the first to offer insecticide-treated eggs of *G. aurantianum* to *T. atopovirilia.* The products that least affected the parasitism were *C. fumosorosea* (3.57%), sulfoxaflor (0%), and cyantraniliprole (0%).

Spinetoram and abamectin most affected the longevity of females, 2.93 and 4.26 days, respectively, compared with 8 days for the control treatment. Similar values were obtained by Takahashi [25] and Pratissoli [34]. Usually, parasitism by *Trichogramma* species peaks in the first five days [14], so if a parasitoid dies before reaching this peak, it would not perform efficiently.

The emergence of individuals from treated *G. aurantianum* eggs was most affected by spinetoram. Khan et al. [26] also found that the emergence of the *Trichogramma chilonis* (Ishii) from eggs of *A. gemmatalis* treated with spinetoram was reduced. These results again show that, regardless of the host, spinetoram causes considerable lethal and sublethal effects on *T. atopovirilia.*

The sex ratio was most affected by cyantraniliprole and cyantraniliprole + abamectin, 0.21 and 0.28, respectively. These numbers are considerably below the expected level for parasitoid-rearing quality [35]. Fertilization of the egg is controlled by the central nervous system [26]. Cyantraniliprole and abamectin block nerve transmissions, which could therefore interfere with the control of fertilization of eggs. This would explain the decrease in emerging females. This hypothesis was also suggested by Delpuech & Meyet [36].

The parasitism of the offspring was higher than their mothers’, and the emergence of F2 was above 80% in all treatments. This can be explained by the pre-imaginal conditioning also found by [28] and [30]. Pre-imaginal conditioning could also be termed “Hopkins’ host selection principle”, i.e., a species that reproduces in two or more hosts will prefer to continue reproducing in the host to which it is most adapted [37].

Spinetoram reduced emergence from *T. atopovirilia* adults by 99.91%. Takahashi [25] found no emergence of *T. atopovirilia* in eggs of *A. gemmatalis* and *C. includens* treated with spinetoram seven days after parasitism. In our study, other products were classified as harmless because the emergence reduction was below 30%. An insecticide may cause mortality in two ways: by contact (penetration into the eggs through aeropiles, reaching the parasitoid) or by ingestion at the egg surface (to emerge, the parasitoid eats the egg corium) [27]. Therefore, spinetoram can cause mortality of the pupa itself and/or prevent the adult from emerging. According to Salgado et al. [38], the lipophilic nature of spinosyns aids in penetrating the egg, but the best activity is observed via injection.

Knowing the persistence of products applied to surfaces can help in practical management by allowing growers to avoid releasing parasitoids until after an insecticide sprayed is no longer toxic to them [2,22,23]. Depending on the period, the parasitoid release can be associated with the application of the active ingredient of interest to improve management. To accomplish this, it is necessary to understand the bioecology of both the biological agent and the pest, to consider the period of greatest efficiency, the peak of parasitism, and the presence of the pest’s eggs [14], for example, respecting the period of action of the product. 

Augmentative releases of *Trichogramma* are now done in Brazil; for some crops, approximately 90–95% of the releases are done with drones at a cost competitive with chemicals [4]. 

This method could also be used in citrus orchards because *T. atopovirilia* can soon be used to control the citrus fruit borer (pers. comm.). Successful use of this egg parasitoid also depends on the compatibility of pest-control techniques, especially biological and chemical tools.

According to our results, the citrus fruit borer could be managed with releases of *T. atopovirilia* at 3–7-day intervals between applications of the active ingredients cyantraniliprole, cyantraniliprole + abamectin, *C. fumosorosea*, sulfoxaflor, and abamectin. Flupyradifurone and spinetoram, on the other hand, should be applied with intervals of 2 or 3 weeks, respectively, between *T. atopovirilia* releases, or could be avoided.

Despite the present results regarding physiological selectivity, the products classified as harmful can still be integrated into management programs, considering their ecological selectivity. In other words, this study provides information to aid in the timing of releases of *T. atopovirilia* to control *G. aurantianum* before, together with, or after insecticides are sprayed, depending on the selectivity and persistence of the insecticides.

## 5. Conclusions

In this study, spinetoram is considered harmful to *T. atopovirilia* and, therefore, should be managed carefully in IPM programs combining this parasitoid. In order to safely use this insecticide, one should respect the interval of the release of the parasitoid, which is 21 days after its spraying. The novel products tested, cyantraniliprole, cyantraniliprole + abamectin, abamectin, sulfoxaflor, and the entomopathogenic fungi *C. fumosorosea* are selective and non-persistent to *T. atopovirilia*. These products are possible replacements for non-selective insecticides to achieve higher control from both chemical and biological tools.

## Figures and Tables

**Figure 1 insects-14-00419-f001:**
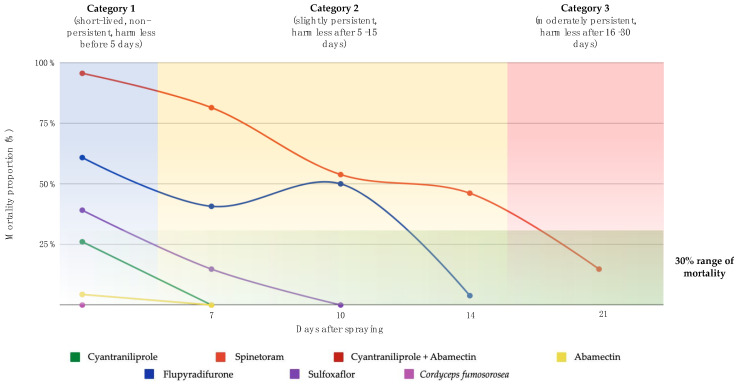
Toxicity and persistence of insecticides on *Trichogramma atopovirilia* adults, based on the IOBC classification. The insecticide cyantraniliprole + abamectin caused the same mortality as Abamectin, so its line is not visible in the figure.

**Table 1 insects-14-00419-t001:** Insecticides evaluated against the parasitoid *Trichogramma atopovirilia*. Standardized commercial product doses for 2000 L of spray.

Active Ingredient (a.i.)	Trade Name, Formulation	Concentration (g a.i. L^−1^)	c.a.i. ^1^	Toxicology Group	Mode of Action
Cyantraniliprole + abamectin	Minecto Pro ^®^, SC	60 + 18	0.40	Diamide +Ivermectin	Ryanodine receptor modulators. Act on the nervous system. Glutamate-gated chloride channel allosteric modulators. Acts on the nervous system and muscle system.
Abamectin	Batent ^®^, EC	18	0.60	Ivermectin	Glutamate-gated chloride channel allosteric modulators. Acts on the nervous system and muscle system.
Spinetoram	Delegate ^®^, WG	250	0.25	Spinosyns	Nicotinic acetylcholine receptor allosteric modulators. Acts on the central nervous system.
Sulfoxaflor	Verter ^®^, SC	240	0.60	Sulfoxamine	Competitive modulators of nicotinic acetylcholine receptors. Act on the nervous system.
Cyantraniliprole	Benevia ^®^, OD	100	0.75	Diamide	Ryanodine receptor modulators. Act on the nervous system.
Flupyradifurone	Sivanto ^®^, SL	200	0.80	Butenolide	Competitive modulators of nicotinic acetylcholine receptors. Act on the nervous system.
*Cordyceps fumosorosea* *	Challenger ^®^, SC	85	0.10	Entomopathogen fungi	Attachment of hyphae or spores. Penetration and internal colonization in the insect body

^1^ c.a.i. (%) = concentration of the active ingredient in the mixture. * *Cordyceps fumosorosea* (Wize) Kepler, B. Shrestha & Spatafora (ESALQ-1296 strain).

**Table 2 insects-14-00419-t002:** Lethal and sublethal effect of insecticides when *Trichogramma atopovirilia* adults are placed in contact with *Gymnandrosoma aurantianum* eggs previously submerged in insecticide solution.

Treatment	Adult Mortality (%)	24 h Parasitism (*G. aurantianum* Treated Eggs) ^1^	Parasitism Reduction (%)	IOBC/WPRS Classification ^2^	Lifespan Parasitism ^1^	Longevity (Days) ^1^
Control	3.33 ± 3.33 a	8.66 ± 1.15 a	--	--	56.00 ± 8.33 a	8.03 ± 0.93 a
Cyantraniliprole + abamectin	10.00 ± 5.57 a	7.60 ± 0.85 a	45.53	2	30.50 ± 4.60 abc	6.36 ± 0.92 bc
Abamectin	10.00 ± 5.57 a	3.71 ± 0.66 b	53.81	2	25.86 ± 5.15 bc	4.26 ± 0.57 c
Spinetoram	46.67 ± 9.24 b	3.79 ± 0.49 b	77.31	2	12.70 ± 6.39 c	2.93 ± 0.72 d
Sulfoxaflor	10.00 ± 4.63 a	5.26 ± 1.02 ab	0	1	60.16 ± 7.86 a	8.47 ± 1.09 a
Cyantraniliprole	3.33 ± 3.33 a	9.36 ± 1.27 a	0	1	62.60 ± 7.11 a	9.20 ±1.00 a
Flupyradifurone	0.00 ± 0.00 a	6.33 ± 1.00 ab	27.62	1	40.53 ± 4.70 ab	7.20 ± 0.71 ab
*Cordyceps fumosorosea*	10.00 ± 5.57 a	6.50 ± 0.79 ab	3.57	1	54.00 ± 8.28 ab	8.50 ± 0.75 a
χ^2^	36.56	26.99			49.91	54.62
*p*	<0.001	<0.001			<0.001	<0.001

^1^ Means ± se followed by the same letter do not statistically differ from each other (Kruskal–Wallis test, followed by Dunn’s test with Bonferroni correction post-hoc (*p* < 0.05)). ^2^ IOBC/WPRS classifications: class 1 = innocuous (<30% reduction), class 2 = slightly toxic (30% to 80% reduction), class 3 = moderately toxic (>80% to 99% reduction), and class 4 = toxic (>99% emergence reduction) [21].

**Table 3 insects-14-00419-t003:** Sublethal effects on the offspring (F1) of *Trichogramma atopovirilia* females exposed to *Gymnandrosoma aurantianum* eggs submerged in insecticide solution.

Treatment	Emergence (%) ^2^	Sex Ratio ^2^	Longevity (Days) ^1^	Parasitism (72 h) ^1^	F2 Emergence (%) ^2,3^
Control	42.42 ± 7.00 ab	0.70 ± 0.04 a	5.41 ± 0.28 a	35.81 ± 5.49 ns	95.22 ± 0.03 a
Cyantraniliprole + Abamectin	46.16 ± 6.37 ab	0.28 ± 0.03 b	3.20 ± 0.48 b	21.00 ± 8.15	86.37 ± 0.09 ab
Abamectin	33.10 ± 10.59 b	0.57 ± 0.13 a	3.44 ± 0.24 b	38.55 ± 6.07	94.09 ± 0.02 ab
Spinetoram	12.03 ± 4.86 c	0.73± 0.14 a	3.84 ± 0.42 ab	27.50 ± 7.44	82.12 ± 0.16 b
Sulfoxaflor	43.76 ± 7.65 ab	0.63 ± 0.05 a	2.50 ± 0.22 b	27.33 ± 7.44	85.46 ± 0.03 b
Cyantraniliprole	28.82 ± 5.79 bc	0.21 ± 0.05 b	3.28 ± 0.47 b	36.85 ± 6.89	95.47 ± 0.01 ab
Flupyradifurone	45.17 ± 7.65 ab	0.68 ± 0.06 a	3.60 ± 0.24 ab	46.80 ± 8.15	91.56 ± 0.02 ab
*Cordyceps fumosorosea*	46.16 ± 6.37 a	0.63 ± 0.10 a	4.25 ± 0.52 ab	36.33 ± 5.51	94.58 ± 0.01 a
χ^2^	27.17	99.89	24.85	9.57	25.83
*p*	<0.001	<0.001	<0.001	0.2146	<0.001

^1^ Means ± se followed by the same letter do not statistically differ from each other (Kruskal–Wallis test, followed by Dunn’s test with Bonferroni correction post-hoc (*p* < 0.05)). ^2^ Means ± se followed by the same letter do not differ statistically (GLM with quasi-binomial distribution, followed by Tukey post-hoc test *(p <* 0.05)). ^3^ F2 generation emerged from parasitized *E. kuehniella* eggs. ns = non-significant.

**Table 4 insects-14-00419-t004:** Effect of insecticides on the emergence of *Trichogramma atopovirilia*, by previous immersion of *Gymnandrosoma aurantianum* eggs containing parasitoid pupae (7 days after parasitism) (mean ± standard error).

Treatment	Emergence (%) ^1^	Parasitism Reduction (%) ^2^	IOBC/WPRS Classification ^3^
Control	76.50 ± 2.67 ab	--	--
Cyantraniliprole + abamectin	74.81 ± 2.78 b	2.75	1
Abamectin	77.82 ± 2.62 ab	0.00	1
Spinetoram	5.42 ± 1.36 c	92.92	3
Sulfoxaflor	80.22 ± 3.15 ab	0.00	1
Cyantraniliprole	74.37 ± 4.10 b	3.21	1
Flupyradifurone	84.28 ± 1.95 a	0.00	1
*Cordyceps fumosorosea*	74.57 ± 2.14 ab	2.20	1
χ^2^	79.75	-	-
*p*	<0.0001	-	-

^1^ Emergence reduction is calculated using the formula R = ((P/*p*) × 100), where R is the percentage of emergence reduction, P is the average value of emergence and *p* is the average emergence in the control treatment. ^2^ Means ± se followed by the same letter do not statistically differ from each other (Kruskal–Wallis test, followed by Dunn’s test with Bonferroni correction post-hoc (*p* < 0.05)). ^3^ IOBC/WPRS classification: class 1 = innocuous (<30% reduction), class 2 = slightly toxic (30–80% reduction, class 3 = moderately toxic (>80–99% reduction) and class 4 = toxic (>99% emergence reduction) [21].

**Table 5 insects-14-00419-t005:** Sublethal effects (after three days of parasitism and emergence) of insecticides on *Trichogramma atopovirilia* in contact with orange leaves treated with insecticides, at 3, 7, and 10 days after application. P = three days of parasitism; E (%) = emergence.

Treatment	Day 3		Day 7		Day 10		Day 14		Day 21	
	P ^1^	E (%) ^2^	P ^1^	E (%) ^2^	P ^1^	E (%) ^2^	P ^1^	E (%) ^2^	P ^1^	E (%) ^2^
Control	22.36 ± 4.11 ab	97.20 ± 1.14 a	58.84 ± 3.44 a	98.65 ± 0.42 a	14.45 ± 3.88 b	99.93 ± 0.06 a	18.23 ± 3.25 ab	100 ± 0.00 a	12.36 ± 1.85	100 ± 0.00 a
Cyantraniliprole + abamectin	24.55 ± 4.07 a	93.28 ± 2.91 a								
Abamectin	9.16 ± 3.74 b	91.92 ± 5.52 a								
Spinetoram	*	*	*	*	*	*	23.00 ± 5.04 a	95.45 ± 3.24 a	7.81 ± 2.19 ns	84.68 ± 4.57 b
Sulfoxaflor	14.67 ± 6.68 ab	88.88 ± 11.11 b	48.17 ± 4.85 ab	93.42 ± 4.25 b						
Cyantraniliprole	26.71 ± 5.68 a	94.86 ± 5.68 a								
Flupyradifurone	33.56 ± 7.72 a	98.36 ± 0.86 a	35.56 ± 5.61 b	96.32 ± 0.75 b	35.56 ± 5.61 a	96.33 ± 0.75 b	9.96 ± 2.72 b	99.27 ± 0.72 a		
*Cordyceps fumosorosea*	31.71 ± 5.29 a	98.09 ± 0.57 a								
χ^2^	21.46	69.55	10.72	18.52	10.14	18.65	8.39	3.89	1.98	20.53
*p*	0.0015	0.008	0.004	<0.001	0.0014	<0.001	0.0151	0.35	0.15	<0.001

^1^ Means ± se followed by the same letter do not statistically differ from each other (Kruskal–Wallis test, followed by Dunn’s test with Bonferroni correction post-hoc (*p* < 0.05)). ^2^ Means ± se followed by the same letter do not statistically differ from each other (Kruskal–Wallis test, followed by post-hoc Tukey test (*p* < 0.05). * On days 3, 7, and 10, there were not enough replicates to analyze. ns = non-significant.

## Data Availability

The datasets generated during and/or analyzed during the current study are available from the corresponding author upon reasonable request.

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
