# Peer review of "Is Integrated Management of Gymnandrosoma aurantianum Possible with Trichogramma atopovirilia and Novel Products Used in Citrus Orchards in Brazil?"

_insects, 2023, doi:10.3390/insects14050419_

Round 1

Reviewer 1 Report

An interesting paper. some details need to be included in the materials and methods and parts of the results need to be rewritten for clarity.

Author Response

All grammatical suggestions were accepted and changed in the manuscript.

ABSTRACT

The term massive implies that the recommended label rates are being far exceeded. I would suggest something like "with multiple insecticide applications, often X in a single season."

Changed for “with multiple insecticide applications, often 56 in a single season”, please see line 25.

Insert a sentence that lists all the products tested. That way, readers don't have to go searching for that information in the materials and methods section.

It was fixed, please see lines 31-33

MATERIAL AND METHODS

Evaluation of lethal and sublethal effects on adults of T. atopovirilia

Where were the experimental arenas kept? What was the temperature range, humidity, and light:dark cycle?

It was fixed, please see lines 161-162

This sentence is confusing. Break it up into several sentences for clarity. “The effects on offspring were evaluated based on the F1 parasitism capacity for three days (number of E. kuehniella eggs parasitized by each 24 hour old female wasp, previously mated, for three days) replacing E. kuehniella eggs daily and the emergence of F2 offspring. The assays of sublethal effects on the offspring were carried out with 10 replicates of each treatment, each female representing one replicate”

It was fixed, please see lines 169-174

Evaluation of lethal and sublethal effects on pupes of T. atopovirilia

Where were the experimental arenas kept? What was the temperature range, humidity, and light:dark cycle?

It was fixed, please see lines 188-189

Residual activity (persistence) of insecticides to T. atopovirilia

These sentences should be added to the end of the last paragraph.If a product reduced insect mortality by less than 30% compared to the control treatment, it was classified according to the IOBC/WPRS persistence scale, as: 1 – short-lived (< 5 days), 2 – slightly persistent (5–15 days), 3 – moderately persistent (16–30 days), and 4 – persistent (> 30 days) [21]. For the surviving females, E. kuehniella eggs were offered and replaced daily for three days to evaluate the parasitism.”

It was fixed, please see lines 210-215

RESULTS        

Following the suggestions all the Tables were fixed

Evaluation of lethal and sublethal effects on adults of T. atopovirilia

The third, fourth, fifth and sixth paragraphs were rewritten following the suggestions. Please see lines 239-256

DISCUSSION

The first two paragraphs were removed/rewritten, please see lines 348-350

Why would they be recommended for IPM if they are selective for non-targets? Shouldn't they be selective for pests?

This sentence was rewritten, please see lines 365-368

Go into more detail. What is the mechanism of exposure on eggs vs. leaves? Why is the persistence test more accurate?

The interaction of the insecticide flupyradifurone on citrus plants may have increased the toxicity of the product, as it is a systemic product, which does not occur with this pest's eggs [42]. Please see lines 392-394

Great statement, but it can't be a paragraph all by itself. Either create a full paragraph or delete this sentence. “Orange growers may better accept a biological pest-control option when selective insecticides are available on the market, as these insecticides facilitate pest management [43].”

This sentence was removed.

Expand the discussion of these results and include information from other research.

This sentence was rewritten, please see lines 415-419

The four lasts paragraphs of discussion were organized, following the suggestions, please see lines 454-470.

CONCLUSION

Following the suggestion, the conclusion was entire rewrite.

Reviewer 2 Report

The authors explored how five new chemical products and one entomopathogen fungi used in citrus orchards may affect adults and pupae of the parasitoid wasp Trichogramma atopovirilia, a potential biological pest-control of the citrus borer Gymnandrosoma aurantianum. To do so, they determined their impacts on parasitism capacity, mortality, and longevity in both F0 and F1 generations. They found that spironetoram is not compatible with T. atopovirilia as it caused significant negative effects on parasitism, longevity, emergence, and mortality while other products had sublethal effects (e.g. sex ratio). The study was well planned, executed, and the data support their conclusions. However, I still have some minor comments and suggestions listed below (order in the text):

 Lines 92-93: It would be informative to have the number and the gps coordinates of the different municipalities.

Line 105: How long does last the development of T. atopovirilia? Please add more information

Lines 184-185: Did you try to transform the data to fit normality of residuals before choosing non parametric tests? I suggest the “bestNormalize” function from “bestNormalize” R package.

Line 189: Please add the alpha level used for statistical analysis

Lines 290-298: The first paragraph of the discussion section sounds redundant with introduction. I suggest to remove it.

Lines 370: please add " by Delpuech & Meyet [45]"

Tables: it would be interesting to use bold font to highlight the significant P values in the different tables

Author Response

Lines 92-93: It would be informative to have the number and the gps coordinates of the different municipalities.

This information has been added. Please se lines 108-110

Line 105: How long does last the development of T. atopovirilia? Please add more information

 This information has been added. Please se lines 121-122

*Lines 184-185: Did you try to transform the data to fit normality of residuals before choosing non parametric tests? I suggest the “bestNormalize” function from “bestNormalize” R package.*

I appreciate your suggestion, but I tried some packages and also transformed the data and it didn't fit, that is, I tried other packages and the data didn't have a normal distribution. Data from counting individuals can hardly be analyzed with parametric statistics.

Line 189: Please add the alpha level used for statistical analysis

This information has been added. Please see lines 221 and 223.

Lines 290-298: The first paragraph of the discussion section sounds redundant with introduction. I suggest to remove it.

 The paragraph was removed/rewritten.

Lines 370: please add " by Delpuech & Meyet [45]"

It was included. Please see line 425.

Tables: it would be interesting to use bold font to highlight the significant P values in the different tables

It was included. Please see Tables.

Round 2

Reviewer 1 Report

Looks good. Just some spelling and grammar changes needed to the additions. 

Author Response

Dear editor, thank you for the comments.   All suggestions were accepted and changed in the manuscript.    
